# Nano-Microbial Remediation of Polluted Soil: A Brief Insight

Shiva Aliyari Rad [1], Khatereh Nobaharan [2], Neda Pashapoor [3], Janhvi Pandey [4], Zahra Dehghanian [1], Venkatramanan Senapathi [5], Tatiana Minkina [6], Wenjie Ren [7,8], Vishnu D. Rajput [6,*] and Behnam Asgari Lajayer [9,*]

1   Department of Biotechnology, Faculty of Agriculture, Azarbaijan Shahid Madani University, Tabriz 3751-71379, Iran
2   12/9 Browns Avenue, Ringwood, Melbourne, VIC 3134, Australia
3   Department of Soil Science, Faculty of Agriculture, Urmia University, Urmia 5756151818, Iran
4   Division of Agronomy & Soil Science, CSIR-Central Institute of Medicinal and Aromatic Plants, Lucknow 226015, Uttar Pradesh, India
5   Department of Disaster Management, Alagappa University, Karaikudi 630003, Tamil Nadu, India
6   Academy of Biology and Biotechnology, Southern Federal University, Rostov-on-Don 344006, Russia
7   Key Laboratory of Soil Environment and Pollution Remediation, Institute of Soil Science, Chinese Academy of Sciences, Nanjing 210008, China
8   College of Resources and Environment, University of Chinese Academy of Sciences, Beijing 100049, China
9   Department of Soil Science, Faculty of Agriculture, University of Tabriz, Tabriz 5166616422, Iran
*   Correspondence: rvishnu@sfedu.ru (V.D.R.); h-asgari@tabrizu.ac.ir (B.A.L.)

**Abstract:** The pollution of soil by heavy metals and organic pollutants has become a significant issue in recent decades. For the last few years, nanobiotechnology has been used to bio-remediate or reclaim soil contaminated with organic and inorganic pollutants. The removal of pollutants from industrial wastes is a major challenge. The utilization of nanomaterials is gaining popularity, which might be accredited to their enhanced physical, chemical, and mechanical qualities. The development of advanced nanobiotechnological techniques involving the use of nanomaterials for the reclamation of polluted soils has indicated promising results and future hope for sustainable agriculture. By manufacturing environment-friendly nanomaterials, the industrial expenditure on decreasing the load of pollution might be reduced. A potential emerging domain of nanotechnology for eco-friendly production and cost reduction is "green biotechnology", alongside the utilization of microorganisms in nanoparticle synthesis.

**Keywords:** bioremediation; environmental pollution; nanomaterials; remediation

## 1. Introduction

Agriculture is considered one of the most important human activities, as it is the main source of food, feed, fuel, and fiber [1]. This activity can cause many environmental problems, especially when insecticides and mineral fertilizers are used in excess [2,3]. Therefore, agricultural contamination might refer to several activities that lead to the destruction or pollution of agroecosystems and affect human well-being [4]. In other words, agricultural soil, soil health, and fertility have been drastically impacted by many different types and classes of pollutants [5]. Some contaminants have a longer lifespan and are recalcitrant. They persist in the soil for many years, disrupting the food chain and causing biological imbalances in the soil, ultimately endangering human health [6]. Pesticides, fertilizers, household and industrial wastewater, industrial activities, and automobile pollution are the major anthropogenic sources of hazardous toxic metals and/or metalloids in the soil [7].

The pressing need is environmental remediation, which must be addressed as a priority [8–10]. In recent decades, various techniques have been used for this purpose, such as mycoremediation [11], phytoremediation [12,13], vermiremediation [4], bioremediation [6],

remediation by using biosorbent materials such as biochar [14], fly ash and organic fertilizers [15], humic substances [16], and nanomaterials (NMs) via green remediation; or combined remediation [17].

The notion of sustainable remediation has recently gained much attention [13], as it essentially aims to reduce the concentration of contamination to risk-free levels while avoiding additional environmental impacts [6]. Several recent developments in this field have combined multiple technologies into a system that provides a cost-effective and time-saving way to disinfect a site while being able to restore the site's quality. As an economical and environmentally sound means of remediating polluted areas, bioremediation is one of the solutions to problems of pollution [18]. The use of microorganisms to remove contaminants from the soil is the key principle of bioremediation [19]. As defined by the Environmental Protection Agency, bioremediation involves the biodegradation of hazardous pollutants to reduce their toxicity or intensity. It offers a number of benefits over physicochemical approaches, including high selectivity, specificity, cost and energy performance, and low demand. However, bioremediation has the disadvantage that it takes longer to degrade toxic compounds, usually several months to a year. It also limits the use of sites that are heavily contaminated with toxic pollutants, resulting in a loss in terms of resource utilization [4,15]. Nanoparticles are used in many scientific fields including automobiles, cosmetics, agriculture, foods, textiles, aviation, defense, engineering, medicine, and the environment [20–22]. According to the National Nanotechnology Initiative of the United States (NNI), there are relatively few studies on using nanotechnology in the analysis and manipulation of materials up to 100 nanometers in size, where unique phenomena enable novel applications of nanotechnology [23]. As an integrated field of nanoscale science, technology, and engineering, nanotechnology consists of viewing, analyzing, modeling, and manipulating materials within this size range. In recent years, nanotechnology has been increasingly used to remove contaminants due to its smaller particulate matter, high surface-to-volume ratio, ease of deployment at impact sites, flexibility, and other advantages [24]. The utilization of nanotechnology for environmental remediation has attracted much attention [23]. Ongoing research and many publications show how nanotechnology can tackle remediation duties and challenges [25,26]. Nanoremediation is a technology that has been recognized as environmentally beneficial by the Environmental Protection Agency. It is acknowledged as a viable strategy for traditional site cleaning [27]. Various techniques for using NMs for soil and water reclamation (nanoremediation) have been reported, such as nano-phytoremediation [28,29], nano-bioremediation [30], nano-$Fe_3O_4$ [31], nano zero-valent iron [32], nano-hydroxyapatite [33], nano zeolite [34], nano zero-valent iron [35], ZnO-nanoparticles (NPs) [36], nano-$TiO_2$ [37], stabilized NPs [38], and nano-silica [39].

This study highlighted the opportunities for contaminant removal, focusing on microbe-mediated remediation and the use of nanomaterials for the reclamation of polluted soils, as well as the benefits and potential risks associated with nanobioremediation technologies. The gaps and future perspectives were comprehensively examined.

## 2. An Evaluation of Nanobioremediation-Based Pollutant Reduction with a Focus on Microbe-Mediated Remediation

Previous techniques for removing heavy metals (HMs) from contaminated soils include biosorption and bioaccumulation utilizing crops and bacteria. However, recent evidence has revealed that the use of NPs in the remediation of HMs has produced impressive results [40]. It has been found that the use of NPs in conjunction with specific microbes, either sequentially or simultaneously, has provided promising results [41]. Not only can they aid in the removal of HMs, but they can also act as nanocarriers for microbial populations or microbial adsorbents [42]. The integration of NPs with microbes for bioremediation is a two-phase procedure that combines biotic and abiotic factors [43]. After entering the system, the contaminants encounter a series of physical methods and revisions that include abiotic mechanisms such as uptake, adsorption, and dissolution, as well as synthetic catalyst supports for photocatalysis during the first stage [44]. Biocides,

bioaccumulation, biostimulation, and biotransformation are examples of biotic systems in the second stage [45]. These biotic systems are essential for removing pollutants from the mechanism. Table 1 provides an overview of various NP-mediated pollutants removed from contaminated media.

**Table 1.** Summary of the various nanoparticle-mediated pollutants removed from contaminated media.

| Nanoparticles | Contaminant Remediated | Factors of Performance and Removal Efficiency | References |
|---|---|---|---|
| Iron oxide nanoparticles with a polyvinyl pyrrolidone coating | Cd, Pb | The use of nanoparticles was combined with a bioremediation process driven by *Halomonas* sp. *Halomonas* sp. was inoculated for 48 h at 180 rpm and 28 °C in the Cd and Pb removal system. After 24 h, 100% removal was detected, whereas it took 48 h for Cd. | [46] |
| Industrial suspension of zero-valent iron (nZVI) at two dosages (1% and 10%) | As | The pH of the nZVI suspension was adjusted to $12.2 \pm 0.1$. Polyacrylic acid was utilized as a stabilizer to prevent the accumulation of nZVI in the suspension. The maximum amount of As immobilized in brownfield soil was 10% of nZVI. | [47] |
| Graphene oxide nanoparticles (nGOx) and nZVI | Metals such as Cd, Pb, Zn, Cu, and As were found in As- and metal-contaminated soil. | The application of nZVI and nGOx to contaminated soils had a significant influence on the availability of As and metals. nGOx immobilized Cu, Pb, and Cd while mobilizing As and P. In the case of nZVI, it successfully immobilized As and Pb (but not Cd) while increasing Cu's availability. This study discovered that both NPs may work as techniques for immobilization and stabilization, which can then be used for phytoremediation. | [48] |
| Titanium oxide nanoparticles bonded to a chitosan nanolayer (NTiO$_2$–NCh) | Cd and Cu | During the experiment, the pH was adjusted at 7.0. The elimination was aided by a microwave-enforced sorption technique that lasted 60–70 s. Cu and Cd were eliminated at a rate of 88.01% and 70.67%, respectively, when NTiO$_2$–NCh was used. | [49] |
| Palladium (Pd), Pd NPs | Cr | Pd NPs were investigated as a bionanocatalyst. Pd NPs were shown to decrease $Cr^{6+}$ completely in 12 h. To decrease 5.0 mol of $Cr^{6+}$, 6.3 mg of Pd NPs was utilized. | [50] |
| Magnetic iron oxide nanoparticles (Fe$_3$O$_4$ NPs) were treated with Staphylococcus aureus and had their surfaces encapsulated in phthalic acid (n-Fe$_3$O$_4$–Phth–S. aureus) | Cu, Ni, Pb | The remediation efficiency of n-Fe$_3$O$_4$–Phth–S. aureus was reported to be 83.0–89.5% for $Cu^{2+}$, 99.4–100% for $Pb^{2+}$, and 92.6–7.5% for $Ni^{2+}$. The researchers also discovered that n-Fe$_3$O$_4$–Phth–S. aureus was an effective biosorbent for removing pollutants. | [51] |
| ZnO NPs | Cu, Cd, Cr, and Pb | The maximum removal of Cr, Cu, and Pb by ZnO-NPs at 5 mg $L^{-1}$ with Bacillus cereus and Lysinibacillus macroides was 60%, 70%, and 85%, respectively. The ideal pH for effective removal was 8.0. The elimination was reduced in the case of bacteria-mediated remediation, which was determined to be 83% and 70% with B. cereus, and 60% and 65% for L. macroides. | [52] |

*Nanobioremediation of Heavy Metals*

Among the most important techniques for removing HMs is site stabilization, which immobilizes the other substances in a particular site, reduces their movement and accessibility in the soil, and prevents them from leaching to other places [53]. The use of numerous NPs, such as biogenic NPs, is becoming popular again for the expulsion of HMs [54]. Biogenic NPs are those produced by biological entities. *Morganella psychrotolerans* produces well-known biogenic NPs, including Ag NPs [55]. FeO nanoparticles produced by coating polyvinylpyrrolidone (PVP) were effectively used with a Gram-negative microbial species, *Halomonas* sp., to improve the bioremediation of pollution created by Pb and Cd. When compared with bacteria or even NPs alone, this method removed nearly 100% of the Pb after 24 h and nearly 100% of the Cd after 48 h [56]. For the removal of Cu, Ni, and Pb, an adsorbent made from permanently magnetized $Fe_3O_4$ NPs was used, which was used with *S. aureus* and also with a powder encased in phthalic acid (as an n-$Fe_3O_4$–Phth–*S. aureus* complex). The precipitation of the adsorbents was 795, 1355, and 985 $\mu mol\ g^{-1}$ for Cu, Pb, and Ni, respectively. In percentage terms, the recovery rates were 83.0–89.5% for $Cu^{2+}$, 99.4–100% for $Pb^{2+}$, and 92.7–7.5% for $Ni^{2+}$. A comparative study with dried S. aureus and n-$Fe_3O_4$–Phth–*S. aureus* for HMs revealed that the n-$Fe_3O_4$–Phth–*S. aureus* core of the NPs played an important role in the excretion of HM in addition to the chemical bonds that existed on the microbial surface [49]. Thus, in this study, the core of the NPs was found to have a significant effect on the excretion of the toxic chemicals in addition to the chemical bonds that exist mainly on the surface of the microorganisms. A recent research on the expulsion of Cu, Cd, Cr, and Pb using HM-resistant bacteria such as *B. cereus* (PMBL-3) and *L. macroides* (PMBL-7) demonstrated that ZnO NPs at 5 mg/L combined with both bacteria removed 60% of the Cr on average, 70% of Cu, and 85% of Pb when compared with *B. cereus* (80% and 60%) and *L. macroides* (55% and 50%) at a neutral pH [48]. The surface of ZnO NPs has negative charges at a neutral pH, which promotes electrostatic attraction to metallic ions. Even so, at low pH values, the HMs occur as hydroxides; thereafter, the hydrogen ions start competing for adhesion with adsorbent materials [57]. The B. cereus strain XMCr-6 has been described as decreasing $Cr^{6+}$ via an enzyme-mediated procedure. The reduced $Cr^{3+}$ covalently attached to the cell lines via coordination bonds with the functional groups on the surface of the bacterial cell membrane. As a byproduct, $Cr_2O_3$ NPs were discovered on the cell membrane [58]. Probiotics (*L. casei* and *L. fermentum*) were also studied for their ability to absorb Cd from water in conjunction with Se5+ and Se NPs. This research revealed greater absorption of Cd by *L. casei* with $Se^{4+}$ ions (65%) in comparison with Se NPs (55.90%), which was directly linked to the higher solubility of $Se^{5+}$, especially in comparison with Se NPs. When *L. fermentum* and *L. casei* were compared, the efficiency of Cd absorption by *L. fermentum* was considerably higher (50.87%) than that by *L. casei* (43.78%). The percentage of Cd adsorbed by *L. casei* combined with Se NPs did not change significantly. Cd absorption slightly increased from 5.49 to 16.54 in the presence of *L. casei* with Se NPs, compared with *L. casei* alone, with increased Se NP ratios [59]. A three-pronged method is gaining some momentum, as HMs pollutants can be utilized by preferential bacteria to produce biogenic NPs (resource recovery), expelling them from the environment (remediation), and producing value from waste (effective waste utilization).

The restoration of HM-contaminated soils is an urgent problem that needs to be solved immediately, both ecologically and in terms of restoring degraded areas [60]. This study discussed the use of plant-based and nanoparticle-based hyperaccumulation systems for the removal of different HMs from polluted areas. The organic materials discovered at a polluted site, as well as the proportion of metal pollutants present, influence how quickly hyperaccumulating plants can be used to remediate the site [61]. The bioavailability of metals in the rhizosphere is determined by the soil's pH, the electrical density gradient, changes in the bacterial community, redox potential, the ratio of $CO_2$ to $O_2$, and other factors [62]. In terms of exudates and root architecture, the rhizosphere also has a direct effect on plant species [61]. Several species of plants, including *Pedioplanis burchelli*, *Amaranthus spinosus*, and *Alternanthera pungens*, can also survive at an effective point of

HMs through rhizofiltration and have displayed HM-avoiding coping processes in their surroundings [63]. HMs have a negative impact on plant development and people's health above an appropriate dose [64]. Nevertheless, hyperaccumulating plant species take up large amounts of metals from contaminated soils, but only after they have transported and accumulated them in larger amounts to organ systems above the soil compared with non-hyperaccumulating plant species, without obvious phytotoxic effects [65]. Plant hyperaccumulators' activity against HMs has been proven by using possible phytoremediation methods including phytostabilization, phytoextraction, and rhizodegradation [66]. The mechanisms of phytostabilization and phytoextraction are responsible for hyperaccumulator plants having a bio-concentration factor (BCF) of more than 1 against HM [62]. A BCF and TF (translocation factor) of more than 1 demonstrates phytostabilization characteristics [67]. Similarly, Kisku et al. [68] discovered the phytostabilization and phytoextraction activities of Parthenium hysterophorus, Sacrum munja, and Ipomoea carnea, and the authors discovered they had a BCF and TF of more than 1 for Cr, Ni, Cd, and Pb, indicating a phytostabilization mechanism, while those with a BCF more than 1 and a TF less than 1 for Zn and Mn indicated a phytoextraction process for HMs. Figure 1 shows a schematic representation of the Me mechanisms of hyperaccumulator plants enriched with NPs to remove toxic compounds from contaminated soils.

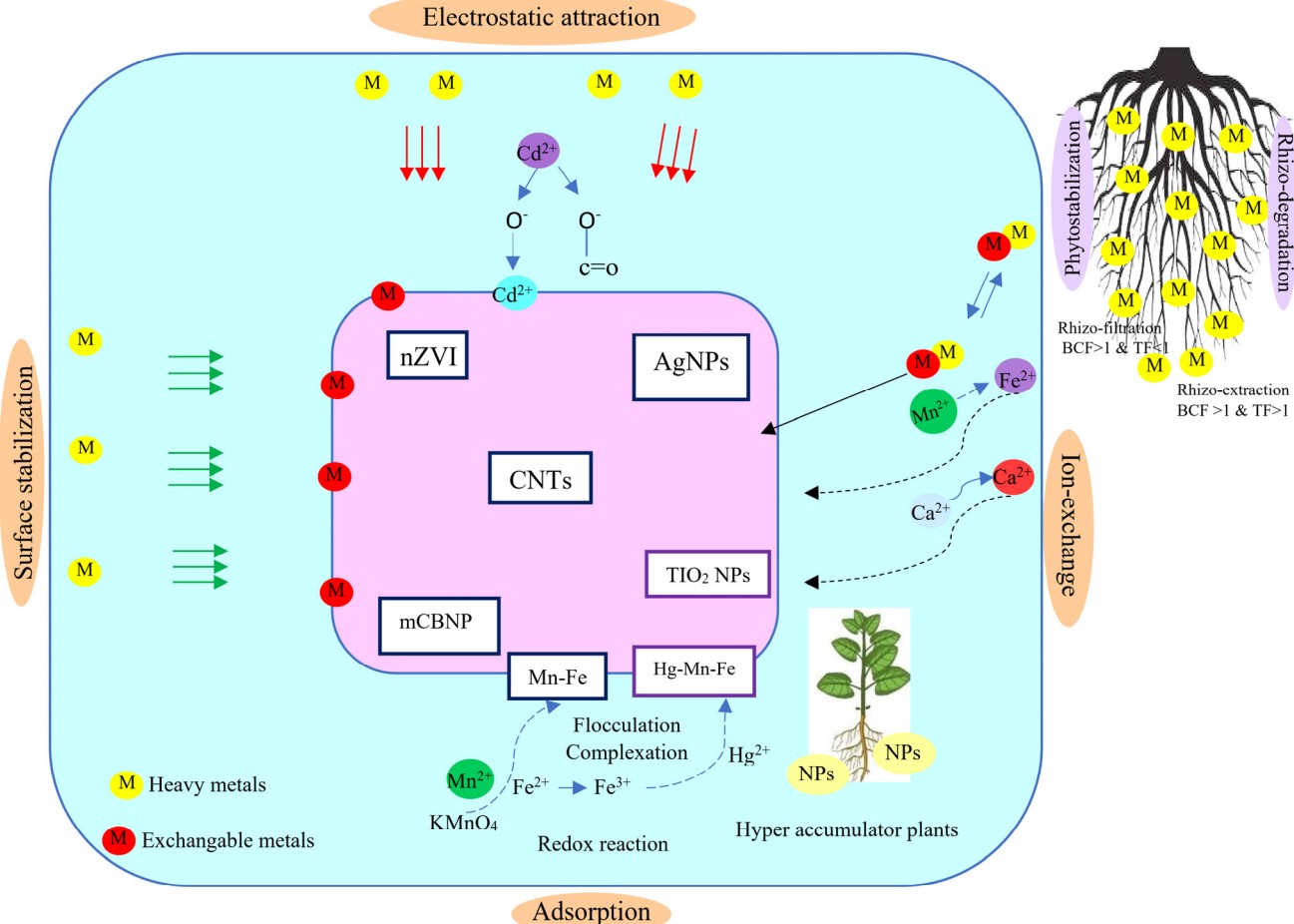

**Figure 1.** Hyperaccumulator plants' mechanism of action supplemented with nanoparticles for removing heavy metals from polluted soils [27].

Rhizodegradation seems to be the method by which pollutants are broken down by bacterial metabolism in the rhizosphere of the soil, where they are metabolized by bacteria for energy and nutrition. Microbes in this system are able to convert harmful

toxins into nontoxic and harmless products [69]. Plant roots release endogenous carbon-containing compounds, including sugar, alcohol, and acid, providing extra nutrients to the microorganisms and stimulating the actions of rhizodegradation [70]. A variety of treatment strategies, such as physico-chemical and biological methods, have been used to decontaminate HM-polluted sites. These methods used redox reactions, adsorbents, ion exchange, bioremediation, and phytoremediation as their processes [71]. Each of these techniques has advantages and disadvantages, and bioremediation has produced much more appropriate eco-friendly methods to achieve long-term goals [72]. Phytoremediation is a widely researched method, and its major applications in polluted soil can be influenced by the addition of components such as NPs. The adsorbent method, among others, plays a critical role in the rapid removal of a broad range of HMs from polluted soils. Adsorbent materials such as activated carbon, biochar, and nanoparticles have recently become widely available. These adsorptive components have rapid adsorption capability, cover a large surface area, provide additional interaction sites for HMs, and are inexpensive [73]. Consequently, the use of hyperaccumulating plants in combination with large adsorbents could be a promising approach to remove HMs from polluted soils. According to the current knowledge, NPs have significant potential for the remediation of HM-contaminated soil. Nanophytoremediation is a method of cleaning up toxins that contain synthetic NPs derived from plants [74]. As a result, ideal methods for removing HMs from polluted soils are needed, such as the choice of hyperaccumulator plants with suitable NPs, which can be an effective method for the remediation of polluted soils.

### 3. Soil Nanoremediation

Nanoremediation is a virtually new application of nanotechnology for addressing environmental pollution issues [75]. Recently, this technique has been used to treat hazardous waste. Although it is a new technical sector, the application of nanotechnologies for environmental remediation has recently attracted a lot of attention from the scientific community [76]. The use of zero-valent iron (ZVI) as a permeability barrier was the first research idea developed by Gillham [77] based on their experience with NPs in decontamination of water-halogenated contaminants [78]. Many researchers utilized chemical synthesis, whereas others use green leaf extracts similar to those used to remove pollutants in aqueous solutions to create zero-valent iron NPs. The use of NPs can effectively degrade numerous pollutants such as organic halocarbons [79], nitrates, HMs [80], pesticides, and dyes [81]. There have been very few studies that have applied NP technology for the remediation of contaminants in soil; studies in this field have instead used for the decontamination of water or aqueous solutions [82].

Studies have reported that NPs can adsorb pollutants and facilitate their destruction through redox reactions, surface reactions, ion exchange, surface complexation, electrostatic contact, and adsorption [83]. A bentonite matrix was used by Shi et al. [82] to remove Cr (VI) from water and soil solutions using ZVI nanoparticles (nZVI) and Fe NPs with zero valency (B-nZVI). They discovered that the use of bentonite (B-nZVI) as a carrier material increased the effectiveness of nZVI nanoparticles, resulting in reduced aggregation and improvements in the active surface area. Likewise, the temperature was directly proportional to the amount of Cr (VI) removed, as were pH and total B-nZVI, which decreased with an increase in pH [84]. B-nZVI NPs have large surface areas and are highly reactive, enabling them to work as excellent adsorbents of Cr (VI) [82]. A wide variety of contaminants have been studied using NPs, including chlorinated organic compounds, insecticides, phenols and amines, organic acids, and chlorinated organic compounds [84]. Two decades ago, experiments showed that NPs, when injected into the soil, could remain effective for up to 56 days and could travel up to 20 m through the groundwater [85]. Zhang [85] reported that it was possible to remove over 99% of trichloroethene (TCE) from polluted locations within a few days.

Studies have shown that zero valent iron NPs trapped in silica microspheres can decompose polybrominated diphenyl ethers, a type of environmental pollutant that can

readily accumulate in the soil [86,87]. Tetrahydrofuran (THF) was used by Qiu et al. [86] to degrade decabromodiphenyl ether from an aqueous solution. The researchers discovered that it was efficient in a solution of THF and water when exposed to environmental and temperature stress. Moreover, the study of Xie et al. [87] suggested that the removal efficiency or elimination efficiency for decabromodiphenyl ether in soil achieved by this degradation process was 78%. It was more significant than the biomass of plants treated with NPs. Additionally, Cr (VI) phytotoxicity was investigated, and iron NPs supported by bio-carbon were tested on cabbage mustard, which showed increased growth and lowered Cr (VI) levels. With the injection of 8 g per kg of soil, the immobilization efficiency for Cr (VI) and overall chromium (Cr) was 100% and 91.94%, correspondingly, in remediation experiments [88]. A lipid derivative of choline-coated silica NPs was used for bioremediation of polycyclic aromatic hydrocarbons (PAHs). Other NMs that have been used include iron sulfide stabilized by carboxymethylcellulose, which was tested for the consolidation of mercury in soils that were heavily contaminated with this metal [89].

Trujillo and Reyes [83] reported the effectiveness of ZVI NPs in remediating aqueous soil solutions contaminated with ibuprofen. They were able to degrade Motrin in aqueous solutions to 54–66% of the original dose, and they achieved an analogous remediation efficiency in sandy soils. Olson et al. [90] used metal NPs (Fe and Mg) to reduce the levels of PCB in soils, achieving a 56% reduction in PCB levels and a $19 \, \text{mg kg}^{-1} \, \text{week}^{-1}$ degradation rate. Additionally, the phytotoxicity test on treated soil samples revealed that almost all PCBs were recovered, as measured by the high germination rate. This type of emulsified ZVI was also used for in situ treatment of polychlorinated soils, where 2-clorobifenilo was completely disrupted.

Other researchers have stressed the importance of further research on soil ecotoxicity and the use of nanotechnology [86], especially with regard to plant-based tests that could be useful as sensitive markers of soil toxicity [85,86]. The mechanisms of NPs' destruction in the soil and their phytotoxicity need to be studied more intensively, especially with regard to plant-based tests that could be useful as a sensitive index of soil toxicity [19]. In 2016, researchers investigated the toxicity of several NMs and their potential to interact in soil. As an illustration, Fan et al. [91] investigated how titanium dioxide ($TiO_2$) influenced the cytotoxicity of copper on Daphnia magna and found that NPs of $TiO_2$ and other pollutants may be hazardous to people if they appear in organic material. Nano-$TiO_2$ exhibited a significant decrease in the toxicity of Cu in D. magna when exposed to humic acid, showing that organic matter in the soil may impair the therapeutic efficiency of such NPs when exposed to specific contaminants. The ability of nano-$TiO_2$ to accumulate HMs depends on the level of adsorption they can achieve. The level of absorption of the metal by nano-$TiO_2$, as well as the presence of humic acids in the solution, affects the ability of nano-$TiO_2$ to accumulate Cu [91].

According to various publications in 2016, the experimental processes and parameters for the synthesis of NPs differ, making it difficult to compare the efficiency gains due to variability in their structures, compositions, and morphologies, all of which impact the adsorption capacities for comparable pollutants. Currently, there is a lack of information on how they break down various types of toxins. The need for comprehensive studies on NMs is underlined by the lack of knowledge on their mechanisms of recovery and reuse, as well as their widespread application and effectiveness for the remediation of industrial effluents and polluted soils. Nevertheless, the reported results have indicated that this remediation technique is valuable compared with conventional techniques.

The effects of nanomaterials on various ecosystems, and their function, life cycle, and release of metal ions are still largely unexplored. Nanoremediation offers several advantages, including lower costs and shorter clean-up times for polluted areas, as well as the possibility to apply it on a large scale. However, to avoid negative impacts on the environment, detailed studies are needed to examine the effects of nanoremediation at the ecosystem level.

## 4. Microorganism-Assisted Nanoremediation

The use of nanoremediation is more sustainable and environmentally friendly if the NPs are biologically produced and microbes are used at the same time. Chemically produced NMs may have many disadvantages in terms of chemical consumption and self-agglomeration in aqueous solutions. In this regard, the utilization of plant extracts, and fungal and bacterial enzymes for green NP production might be a promising option. In this process, metallic NPs are created due to their reducing effect on the metal complex salts. Co-precipitation, or the addition of proteins and bioactive components to the outer surfaces of the NPs, greatly increased their strength in an aqueous environment. Mahanty et al. [92] found Aspergillus tubingensis (STSP 25) biofabricated iron oxide NPs from the rhizosphere of *Avicennia officinalis* in Sundarbans, India. About 90% of the HMs (Ni (II), Cu (II), Pb (II), and Zn (II)) in wastewater were eliminated or removed by the synthesized NPs, which had a regeneration potential of up to five cycles. The metal ions were chemically bound to the surface of the NPs by an endothermic reaction [92]. The co-precipitation of iron oxide NPs and exopolysaccharides (EPS) from Chlorella vulgaris has been described in other studies. The effective alteration of NPs by EPS functional groups was demonstrated by FT-IR spectroscopy. It was also demonstrated that the nanocomposite could remove 85% of $NH_4^+$ ions and 91% of $PO_4^{3-}$ ions [93].

It has been claimed that using bacteria to produce NPs is a practical and beneficial method for the environment. A copper-resistant Escherichia species, SINT7, was used to synthesize copper NPs. Biogenic NPs were observed to degrade azo dyes and textile effluents. Consequently, at a lower concentration of 25 mg/L, 83.6%, 90.6%, 97.1%, and 88.4% of reaction black-5, malachite green, Congo red, and direct blue-1 were lowered, respectively. When the concentration was increased to 100 mg/L, they reduced by 76.84%, 31.1%, 83.90%, and 62.32%, respectively. Additionally, treated samples of industrial sewage contained less phosphate and chloride ions, along with the suspended particles. The performance of biogenic NPs such as these may boost cost-effective and long-term industrial manufacturing [94]. Cheng et al. [95] used no additional sulfur to make iron-sulfur NPs. These NPs had the ability to annihilate Naphthol Green B dye through the extracellular transfer of electrons. The utilization of *Pseudoalteromonas* sp. CF10-13 in manufacturing NPs offers an environmentally acceptable biodegradation method. The manufacturing of toxic gases and metal complexes was constrained by the endogenous creation of NPs.

The use of biological particles is a more effective way to remediate industrial wastewater. As well as the direct production of NPs from microbes, there are several other ways in which microorganisms can contribute to the advancement of nanotechnology. In addition to NPs, microorganisms can also provide catalytic enzymes that help in wastewater treatment. Table 2 gives a quick overview of the application of nanotechnology in the bioremediation of wastewater.

**Table 2.** Utilization of advanced nanotechnology mechanisms for the bioremediation of various industrial effluents.

| Nanotechnology Administered | Modification | Affiliated Microorganisms | Deletion or Adsorption Efficiency | Advantage/Technique | Distinct Attribute | References |
|---|---|---|---|---|---|---|
| NiO and MgO nanoparticles | SiO$_2$ embedding | - | Maximal absorption at a rate of 41.4, 13.8, and 7.2 (ions/nm$^2$) for Cr$^{3+}$, Cu$^{2+}$, and Zn$^{2+}$, respectively | Physical, spontaneous, and endothermic absorption of Cu$^{2+}$ and Cr$^{3+}$, but chemical and exothermic Zn$^{2+}$ uptake | Renewal, reusability, and proven sustainability | [96] |
| Electrospun nanofibrous webs | Bacterial encapsulation | *Pseudomonas aeruginosa* | 55–70% deletion of methylene blue at various concentrations. | Biological deletion of dye | Potent bacterial cells or genetic engineering can be rather promising. | [97] |
| Mesoporous organosilica nanoparticles (MONs) | Ferrocene amalgamation | - | Application of MONs increased the removal rates of dyes by 50% and metals by 25% | Ferrocene facilitated the non-covalent interaction and provided a larger surface area and conjugation | Advanced organic–inorganic hybrid nanomaterial | [98] |
| Co and CoO nanoparticles | Microwave and reductive chemical heating | - | Respectively, cobalt and cobalt oxide nanoparticles destroyed murexide dye by 43.6 and 39.4%. | Irradiation and greater surface area | Eco-friendly, easy to build, fast, and highly efficient photocatalytic degradation | [99] |
| Electrospun cyclodextrin fibers | Bacterial encapsulation | *Lysinibacillus* sp. | Reduction efficacy: Cr(VI) = 58 ± 1.4%; reaction black 5 = 82 ± 0.8%; Ni(II) = 70 ± 0.2% | Bacterial bioremediation | Cyclodextrin contributes to bacterial growth by providing an additional carbon source | [100] |
| Zirconia (ZrO$_2$) nanoparticles | Synthesis based on a microbial acellular culture supernatant | *Pseudomonas aeruginosa* | Tetracycline accumulation up to a concentration of 526.32 mg/g | Chemisorption and potent electrostatic reaction amongst zwitterions | Synthesis of green nanoparticles and steady bioremediation | [101] |
| Enzyme-immobilized nanoparticles | Laccase immobilization | *P. ostreatus* | Breakdown of 90% bisphenol-A and 10% carbamazepine | Immobilized laccase-mediated oxidation | Reusable and cost-effective enzyme | [102] |
| Graphene oxide (GO) and carbon nanotubes | Nanosized Ni metal–organic framework | - | Methylene blue accumulation up to 222 mg/g | Mixed nanocomposites consist of hydrophobic and/or π-π interactions, a large surface area, pores between MOFs, and diverse morphological characteristics. | The nanocomposite's interaction was far better. | [103] |
| Silica (SiO$_2$) nanoparticles | Synthesized from actinomycetes | *Actinomycetes* | Approx. 80% clearing of industrial wastewater | Photocatalytic deterioration | Cost-effective and stable | [104] |

## 5. Utilization of Nanomaterials for Micro-Remediation of Polluted Soils

Bioremediation using of microorganisms has been proposed as a supposedly efficient approach to remediating contaminated sites [105]. Microorganisms that are capable of modifying soils contaminated with HMs and organic pollutants have attracted much attention. Volatilization, metal-binding, alteration, and chemical precipitation are some of the techniques used for the remediation of HMs using microorganisms [106,107]. According to Xu et al. [108], the following elements bind metals to microbial cells: $CrO_4^{2-}$, $Cu^{2+}$, $Hg^{2+}$, $Au^{3+}$, $Cd^{2+}$, $Ni^{2+}$, $Pd^{2+}$, and $Zn^{2+}$. The mobility of these metals and their harmful consequences were diminished by this metal-binding. Furthermore, Polti et al. [109] studied the use of microorganisms for the bioremediation of Cr (VI)-contaminated soils. Soil samples showed that the Streptomyces species MC1 was able to reduce Cr (VI) to Cr (III), the latter being more stable and less hazardous than the former. Metals that are volatile, such as Hg, can be volatilized by microbes [110]. On the other hand, organic pollutants can be destroyed by a variety of microorganisms or enzymes. Certain microorganisms can use the nitrogen and carbon in organic contaminants, leading to soil decontamination.

For example, four microbes were isolated from soils planted with bamboo, pine, and rice to treat polluted soils, including *Rhodotorula glutinis* 4CD4, *Pseudomonas nitroreducens* 4CD2, *Pseudomonas putida* 4CD1, and *Pseudomonas putida* 4CD3. All the isolated microorganisms effectively broke down p-hydroxybenzoic acid, ferulic acid, p-coumaric acid, and p-hydroxybenzaldehyde using phenols as a carbon source [111]. *Pseudomonas stutzeri* OX1 has also been shown to be able to break down tetrachloroethylene. Due to the production of toluene-xylene monooxygenase, which induces the aerobic breakdown of pollutants in bacteria, researchers ascribed this degradation to *Pseudomonas stutzeri* OX1 [112]. The effectiveness of micro-remediation for remediating pollutants is impacted by the potential impact of NMs on microorganisms. Shrestha et al. [113] investigated the influence of NMs on the architecture and function of soil microbial communities using MWCNTs. According to pyrosequencing research, applying $10$ g kg$^{-1}$ MWCNTs increased the abundance of many bacterial taxa such as *Cellulomonas, Pseudomonas, Nocardioides*, and *Rhodococcus*, which are thought to be potential degraders of resistant pollutants. NMs were also observed to affect the level of microbial assembly in the soil in favor of species that were more resilient to NMs or were capable of rapid degradation, which was advantageous for soil micro-remediation. Research on the breakdown of 2,4-dichlorophenoxyacetic acid in soils was carried out utilizing $Fe_3O_4$ NPs in combination with soil microorganisms. The addition of $Fe_3O_4$ NPs to the soil increased the microbial diversity and enzymatic activity (e.g., acid phosphatase, amylase, urease, and catalase), resulting in greater organic waste degradation efficiency than when the soil was treated with microorganisms alone [114]. Tilston et al. [115] discovered that the use of nZVI coated with polyacrylic acid (PAA) altered the composition of the bacterial community in the contaminated soil and reduced the efficiency of chloroaromatic mineralizing microorganisms. Populations of Dehalococcoides, a bacterium capable of dechlorinating chlorinated organic pollutants, were similarly reduced with $0.1$ g L$^{-1}$ nZVI [111].

The adverse upper effect of NMs on microorganisms prevented the biodegradation of pollutants in polluted soils. Furthermore, the impact of NMs on micro-remediation of polluted soils differs depending on the type and concentration of NMs. With increased CNT concentrations, extractability and microbial degradation of PAHs were assumed to decrease with an increasing CNT concentration. Compared with MWCNTs, SWCNTs had a stronger influence on the mineralization and extraction of PAHs [116]. Furthermore, a high concentration of MWCNTs significantly inhibited the development of phenanthrene-catabolizing bacteria as well as the growth of phenanthrene-degrading bacteria in the soil, while fullerene and low levels of CNTs had no negative influence on microbial activity [117]. When discussing bioremediation, it is important to remember that the pollutants that NMs ingest affect their bioavailability to microorganisms. The reduced bioavailability of contaminants affects the microbial remediation power in the polluted soils. MWCNTs

adsorbed on phenanthrene were studied for their biodegradation and mineralization by *Agrobacterium*. It was found that the use of MWCNTs as contaminants significantly reduced the bioavailability of hydrophobic organic compounds in the environment [118]. Other research used 14C-2,4-DCP as the target contaminant to explore the mineralization, breakdown, and residual distribution of radioactively tagged 2,4-dichlorophenol (14C-2,4-DCP) in conjunction with SWCNTs and MWCNTs. In contaminated soils, SWCNTs at a concentration of $2\,g\,kg^{-1}$ significantly reduced microbial mineralization and the breakdown of 14C-2,4-DCP. The reduced bioavailability of 2,4-DCP, the potential microbial toxicity of CNTs, and the reduced activity of native soil microorganisms all had inhibitory effects [119]. Overall, NMs have both beneficial and detrimental effects on the micro-remediation of contaminated soils (Figure 2).

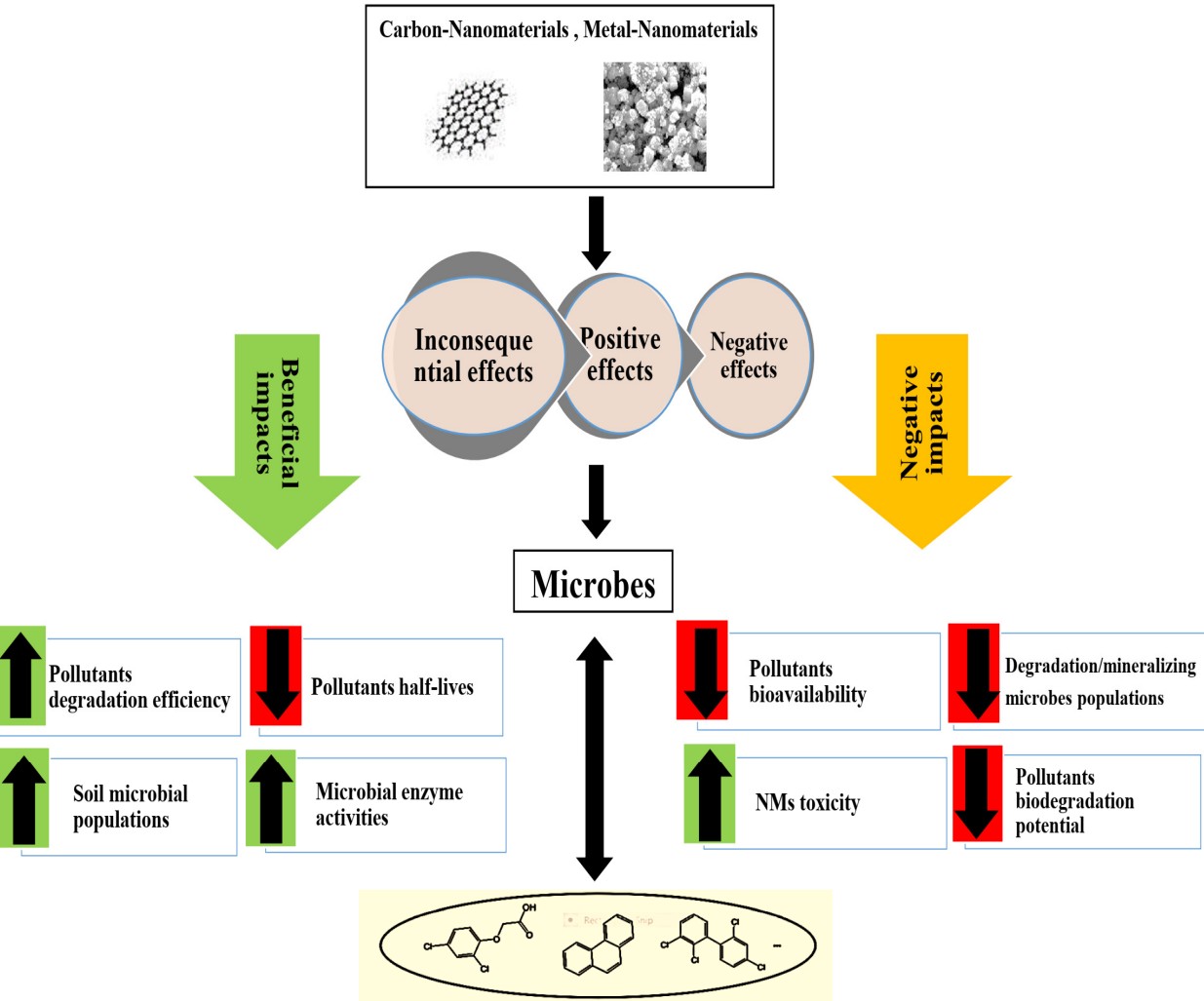

**Figure 2.** Beneficial and negative impacts of nanomaterials on micro-remediation of soil. Green arrows indicate upregulation. Red arrows indicate downregulation.

## 6. Advancements in Agricultural Techniques by Using Nanotechnology

Nanotechnology is a rapidly growing field of science that focuses on the design, characterization, production, and application of structures and components. Nanotechnology can be used to create structures, components, or systems that may have a new feature or provide a better asset by changing their size and shape at the nanoscale level (atoms and molecules smaller than 100 nm) [120]. These novel substances are intentionally designed to offer advantages over traditional substances. As a result, NPs have achieved large-scale of production, estimated to be about 260,000–309,000 metric tons in 2010 [121].

In order to find sustainable and environmentally friendly solutions, 10 years of research have been conducted, according to a 2013 FAO report on the state of nanotechnology in food and agriculture [122]. For example, research has focused on the intellectual release of active constituents (disease management and crop defense), reducing nutrient losses during fertilization while enhancing performance, and creating bio–nano combinations from traditional crops [123]. By 2050, the population will grow to 9.9 billion people, requiring an increase in food production [122]. A growing population will require the optimization of resources (soil, water and inputs) and the restoration of degraded soils for agricultural use. As oil resources decline, there will be a significant shift to energy crops. According to the FAO's 2009 expert discussion, "How to Feed the World in 2050", cereal production will need to increase by 70% by 2050. Nanotechnology could open up new opportunities for modern agriculture and help to solve the challenges of future food and energy needs in a stable way. Our understanding of the interactions of ENM with soil is limited. Due to the complexity of these systems, we still have a long way to go before we fully understand the behavior of anthropogenic NPs. In terms of cutting-edge issues such as improving fertility, reducing erosion, mitigating or degrading pollutants, and creating nutrient and pollutant sensors, the last 15 years of soil research have been positive [124].

In the construction of nanocomposites and nano-capsules, the content of the active ingredient should be properly absorbed during cultivation to avoid overdosing and to reduce inputs and wastage. Rationalizing and controlling the rate of application of fertilizers [121,122], herbicides [123,124], pesticides [125], or developmental enhancers [126] can reduce the overall cost of remediating potentially polluted land [127]. Numerous assessments on the use of potentially valuable NMs for the recovery and improvement of soil have been published [128,129]. Nano-fertilizers have enabled nutrients' translocation in the rhizosphere, improving the composition and dosage of the fertilizer applied. It is now known that plants consume only a limited amount of fertilizer, the rest of which is lost during processing, leaching, and mineral retention. The use of nanoscale vehicles or NPs that penetrate the root and enhance the absorption of beneficial compounds is currently being explored. These vehicles can also be used to detect and bind hazardous soil particles.

To reduce nutrient loss, Hussein et al. [125] used a composite material based on zinc and aluminum hydroxide as a cover. Kottegoda et al. [126] used wood to enclose urea hydroxyapatite NPs, which released nitrogen after 60 days vs. 30 days with fertilizer. There is a list of registered products, one of which is for grain crops and includes a mixture of macronutrients (nitrogen, phosphorus, and potassium), micronutrients, mannose, and amino acids [127]. The loss of nitrogen via leaching, which causes the eutrophication of water, is well known. Composite NMs (mixtures of plastic and starch) were used to coat fertilizers, helping to solve these problems [128]. The water retention of NMs has been extended in nanoclays, nanozeolites, and nanohydrogels [129]. Sekhon [130] provided a thorough examination of these and their novel qualities. When covered with zinc, nanoclays composed of polyacrylamide polymers have a high water absorption capacity and water content. This is a crucial feature, as they can be used on rain-fed crops [131]. Mahfoudhi and Bouf [132] designed nano-hydrogels based on cellulose nano-fibrils (CNFs) and polyacrylic acid–co-acrylamide. The system released urea through the established structure, which mimicked a fertilizer. The combined capabilities of NMs to improve soil conditions were shown in recent research. Kottegoda et al. [126] designed hydroxyapatite NPs (derived from $H_3PO_4$) and inserted them in gaps of clay platelets and achieved a gradual phosphate discharge; the same researchers also inserted altered cellulose for the same protective action. Liu and Lal [133] created synthetic apatite NPs gradually loaded with phosphorus for soybean plants (*Glycine max* L.).

Pesticides are loaded with mesoporous silica (MSN), which protects the active ingredient (avermectin) from photodegradation and are dosed slowly so that they remain active for a long time [134]. Due to their biodegradability, they are effective at crop protection and eventually become "soil friendly". Khot et al. [135] used two actives (nano-imidacloprid in conjunction with $Ag/TiO_2$) encapsulated inside a mixture of chitosan and alginate for

disease suppression. Because of this, the remainder of the formulation had decomposed in the soil after 8 days. A comprehensive review on bio-nanocomposites was provided by Zhao et al. [136], which featured the benefits of using a polymeric matrix derived from proteins or starch to preserve fertilizer and to produce nanobioplastics. The utilization of biochar has displayed intriguing results, such as soil amendment and the absorption of a variety of unfavorable residues [5]. When compared with conventional sensors, nanosensors provide more advancements and enhanced functionality. Nanosensors are designed to evaluate dimensions of less than 100 nanometers. In response to the introduction of other composites of a comparable size, nanotubes, NPs, nanocrystals, and nanowires transmit an electromagnetic signal. Installing nanosensors in fields can allow farmers to monitor the soil conditions in real time and recognize problems such as water deficits and soil nutrient demands early. Nanomaterials have a significant surface response, providing a quick response to identify environmental conditions better. This could be a useful addition to smart agriculture in the coming years to find better solutions to agronomic problems [137].

## 7. Conclusions and Future Prospects

The transformation and detoxification of many environmental pollutants from the soil could be achieved through nanoremediation using NMs. The use of NMs can enhance the effects of bioremediation by increasing the uptake and accumulation of pollutants in plants and increasing the rate of pollutant degradation by microorganisms. The negative effect of NM on organisms inevitably has serious implications for the bioremediation process in soil. To make soil remediation more effective, NMs need to be carefully studied in combination with biotechnology to determine how they interact with plants/microbes in polluted soils, their likely fate, and whether NMs affect pollution. As soil bioremediation is still in its infancy, more extensive studies are needed before NMs can be used to aid the process. The original advantages of using nanomaterials in soil remediation are the decrease in the remediation time and the overall cost, the reduction of pollution to near zero within the site, and the fact that no disposal of the contaminated soil is required. Due to their tremendous reactivity and strong ability to immobilize HMs such as Cd, Ni, and Pb, nZVI nanomaterials are widely used for environmental remediation. It is possible to reduce the toxic effects on soil microorganisms by modifying and/or capping nZVI. Due to their large surface area and high adsorption capacity, carbon nanotubes (CNTs) are excellent nanomaterials for organic and inorganic remediation. To better understand how CNTs affect the environment, further research is needed.

**Author Contributions:** All authors (S.A.R., K.N., N.P., J.P., Z.D., V.S., T.M., W.R., V.D.R. and B.A.L.) contributed equally to writing part of the original draft, and also reviewed and edited the whole manuscript. All authors have read and agreed to the published version of the manuscript.

**Funding:** This research received no external funding.

**Institutional Review Board Statement:** Not applicable.

**Informed Consent Statement:** Not applicable.

**Data Availability Statement:** All data, tables, figures, and results in this study are our own and original.

**Acknowledgments:** V.D.R. and T.M. acknowledge support by the laboratory "Soil Health" of the Southern Federal University and the financial support of the Ministry of Science and Higher Education of the Russian Federation (agreement No. 075-15-2022-1122).

**Conflicts of Interest:** The authors declare no conflict of interest.

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
