# Peer review of "Nano-Microbial Remediation of Polluted Soil: A Brief Insight"

_sustainability, doi:10.3390/su15010876_

Round 1
Reviewer 1 Report
The manuscript sustainability-2009754 “Nano-microbial remediation of polluted soil: A brief insight” is an interesting review article with a good revision and synthesis of current literature referred to nanotechnology applications. The article shows new insights related to the remediation and reclamation of different environmental matrix by the utilization of nanoparticles and nanoparticles in combination with different materials and microorganisms. Despite the interesting topic covered in this review, there are several shortcomings that need to be improved before to be considered for published in the special issue and journal.
The article reviews several articles related with the thematic proposed, nevertheless and despite the short extension of the manuscript, in some parts of the text it can be confused for the readers. In this sense, format and acronyms need to be enhanced and well used in the manuscript, with the corresponding definition, some concepts also must be avoided and refer to the correct one. Moreover, there are several references that could be presented in other tables, and figures would be better.
In terms of content, the article should start with some elemental definitions for enhance the understanding of the readers, such as the type and origin of NP revised, the type of pollutants and its main ecotoxicological effect on environmental matrices and some concept like availability-bioavailability among others. In this sense, the manuscript is very poor and confuse. In this context, some mechanisms related to the action of NM must be presented initially and subsections for the different types of pollutants in should be also included.
Moreover, when refer to the different environmental matrixes and organisms detailed information is mandatory. For example, soil matrices and types are significantly consistently different in composition, physicochemical and biological properties, sorption characteristics among other, thus, these antecedents must be also presented when explanation and statements are described for the reported effects of NP. Therefore, and considering structure and contents of the article I suggest handling back the manuscript to the authors with major revisions for being considered for publication.
Author Response
We are highly thankful for your valuable comments and suggestions. Please see the attachment.

Reviewer 2 Report
Reviewer's comments
The review paper is good it has good contents but it is not properly organized. Some of my comments are given on the manuscript attached.
But generally, the authors need to organize their work properly, in some part they misused thee wrong terminology and confusing language. For example the authors used degradation of contaminants for both heavy metals and organic pollutants. Heavy metal don't degrade , degradation is only for organic pollutants . They also miss use several terminologies . The authors are advised to classify the contaminants as organic pollutants and heavy metal pollutants and they can distinctively write their reviews with no confusion. The authors need to match their subtopic with the connets of the subtopic . For example in section 4 , they say nanotechnology assisted remediation , but they wrote about microorganism at the bottom. Thus all in all the contents are very interesting but the pollutants need to be classified and the subsections should clearly reflect the subtopic. The authors english is very poorly written , it is very hard to understand what they mean mostly or use the wrong word. Thus they should get it edited by native speakers or fluent English writers. The authors in some parts mixed between nanotechnology and nanoparticles, they need to revise the whole manuscript clearly and adequately.

Author Response

(The authors gave the same response as above.)

Reviewer 3 Report
Detailed comments are as follows:
1. Please, the keywords section should start with the "texture".
2.Where is the practical application of this manuscript? It must be added.
3. Check the grammar throughout the article and correct it. Proofread the article as many language errors were identified.
4. The introduction needs to be further revised to highlight the purpose of the study, You need to introduce what others have studied and what needs further research. Besides, the following all of references are recommended to be cited:
https://www.sciencedirect.com/science/article/abs/pii/S0360319922008126
https://www.sciencedirect.com/science/article/pii/S0167732221011296
5. The conclusion must be more than just a summary of the manuscript.
Author Response

(The authors gave the same response as above.)

Round 2
Reviewer 2 Report
The authors have made significant improvements of the manuscript, I suggest a minor revision of the manuscript with some of my comments written on the manuscript attached

Author Response
We are highly thankful for your valuable comments and suggestions.

Reviewer 3 Report
Accept
Author Response
We are highly thankful for your valuable comments and suggestions
